# Effectiveness of orally-delivered double-stranded RNA on gene silencing in the stinkbug *Plautia stali*

Yudai Nishide[1]*, Daisuke Kageyama[1], Yoshiaki Tanaka[1], Kakeru Yokoi[1], Akiya Jouraku[1], Ryo Futahashi[2], Takema Fukatsu[2,3,4]*

1 National Agriculture and Food Research Organization (NARO), Institute of Agrobiological Sciences Ohwashi, Tsukuba, Japan, 2 National Institute of Advanced Industrial Science and Technology (AIST), Tsukuba, Japan, 3 Department of Biological Sciences, Graduate School of Science, The University of Tokyo, Tokyo, Japan, 4 Graduate School of Life and Environmental Sciences, University of Tsukuba, Tsukuba, Japan

* nishiyu0@affrc.go.jp (YN); t-fukatsu@aist.go.jp (TF)

**Data Availability Statement:** The data underlying this study are available on Dryad (https://doi.org/10.5061/dryad.x3ffbg7h2).

## Abstract

Development of a reliable method for RNA interference (RNAi) by orally-delivered double-stranded RNA (dsRNA) is potentially promising for crop protection. Considering that RNAi efficiency considerably varies among different insect species, it is important to seek for the practical conditions under which dsRNA-mediated RNAi effectively works against each pest insect. Here we investigated RNAi efficiency in the brown-winged green stinkbug *Plautia stali*, which is notorious for infesting various fruits and crop plants. Microinjection of dsRNA into *P. stali* revealed high RNAi efficiency–injection of only 30 ng dsRNA into last-instar nymphs was sufficient to knockdown target genes as manifested by their phenotypes, and injection of 300 ng dsRNA suppressed the gene expression levels by 80% to 99.9%. Knock-down experiments by dsRNA injection showed that multicopper oxidase 2 (MCO2), vacuolar ATPase (vATPase), inhibitor of apoptosis (IAP), and vacuolar-sorting protein Snf7 are essential for survival of *P. stali*, as has been demonstrated in other insects. By contrast, *P. stali* exhibited very low RNAi efficiency when dsRNA was orally administered. When 1000 ng/µL of dsRNA solution was orally provided to first-instar nymphs, no obvious phenotypes were observed. Consistent with this, RT-qPCR showed that the gene expression levels were not affected. A higher concentration of dsRNA (5000 ng/µL) induced mortality in some cohorts, and the gene expression levels were reduced to nearly 50%. Simultaneous oral administration of dsRNA against potential RNAi blocker genes did not improve the RNAi efficiency of the target genes. In conclusion, *P. stali* shows high sensitivity to RNAi with injected dsRNA but, unlike the allied pest stinkbugs *Halyomorpha halys* and *Nezara viridula*, very low sensitivity to RNAi with orally-delivered dsRNA, which highlights the varied sensitivity to RNAi across different species and limits the applicability of the molecular tool for controlling this specific insect pest.

**Funding:** This study was supported by the JSPS KAKENHI Grant JP16K21613 and JP19K06080 to Y.N. and the JST ERATO grant JPMJER1803 and JPMJER1902 to T.F. The funders had no role in study design, data collection and analysis, decision to publish, or preparation of the manuscript.

## Introduction

Since its discovery at the end of the previous century, gene silencing mediated by administration of double-stranded RNA (dsRNA), called RNA interference (RNAi), has been a powerful technique for unveiling gene functions. Following the discovery of RNAi in the nematode *Caenorhabditis elegans* [1], high sensitivity to RNAi was found in coleopteran insects such as the red flour beetle *Tribolium castaneum* [2, 3], the Colorado potato beetle *Leptinotarsa decemlineata* [4], and the western corn rootworm *Diabrotica virgifera* [5]. By contrast, lepidopteran, dipteran and Odonata insects were found to be less sensitive to RNAi [6–8]. Previous RNAi studies have uncovered a substantial amount of variability in RNAi sensitivity among different insect species and lineages, but it is obscure what relationship exists between RNAi sensitivity and insect taxa.

Conventionally, RNAi experiments have relied on dsRNA injection into insect bodies because of generally efficient gene silencing by this method. However, injection entails several problems including immune and wound stresses especially in small insects. By contrast, oral delivery of dsRNA is less traumatic and easier to perform because no special equipment is needed. Additionally, RNAi by feeding is practically applicable to crop protection [5, 9], which has become one of the promising techniques for pest management over the past decade. Here, a critical challenge in developing insect-specific molecular biopesticides is to find effective and reliable methods for delivery of dsRNA into pest insect bodies. Arming plants with dsRNA has been suggested [10], but all such methods hinge on the idea that crop is deployed with the corresponding dsRNA of essential insect genes. This approach could reduce our dependence on chemical insecticides and could combat insect resistance to chemical insecticides [11, 12]. An example of deploying dsRNA to plants is provided by transgenic maize to control the western corn rootworm *D. virgifera* [5, 13, 14]. RNAi by feeding mediated by transgenic rice that produces dsRNA was tested and successfully inhibited gene expression in the brown planthopper *Nilaparvata lugens*, although lethal phenotypes were not observed [15, 16]. Because RNAi exhibits species specificity, fewer concerns arise for off-target effects, an important issue in crop protection [17–19]. The efficiency of RNAi delivered by feeding varies considerably among different species including hemipterans [20].

In this study, we tested the efficiency of RNAi by injection and feeding of dsRNA in the brown-winged green stinkbug *Plautia stali* (Hemiptera: Pentatomidae). Previous studies reported that dsRNA injection efficiently suppresses gene expression in this species [21–23], and maternal RNAi also works effectively [24]. However, the effect of orally-delivered dsRNA is unknown. *P. stali* has attracted much attention in the field of microbiology and evolutionary biology regarding insect–symbiont interactions [25–30]. Development of a reliable method for RNAi by feeding will accelerate the functional studies on symbiosis-related genes even in very tiny first-instar nymphs of *P. stali*, in which symbiont transmission and establishment occur [26, 30]. Furthermore, because *P. stali* is a notorious agricultural pest infesting various fruits and crop plants [31], RNAi by feeding would provide a useful method for control and management of *P. stali* and other pest stinkbugs.

## Materials and methods

### Insects

Adult insects of *P. stali* were collected at a forest edge in the National Institute of Advanced Industrial Science and Technology, Tsukuba, Japan, from which an inbred laboratory strain was established. A mass-reared colony of the strain was used as the source of experimental insects. The insects were reared in plastic containers (150 mm in diameter, 60 mm high) with

raw peanuts, dry soybeans, and drinking water supplemented with 0.05% ascorbic acid at 25˚C ± 1˚C under a long-day regime of 16 h light and 8 h dark as previously described [29].

## Target genes

RNAi efficiencies are often highly variable across different target genes [32, 33]. In this study, we selected four target genes: vacuolar ATPase (vATPase), inhibitor of apoptosis (IAP), multi-copper oxidase 2 (MCO2), and vacuolar-sorting protein Snf7. vATPase and IAP were reported to cause high mortalities upon effective RNAi in several hemipteran insects [17, 34–41], and therefore, they are regarded as promising targets for pest control. For vATPase gene, we performed RNAi using three distinct dsRNAs designed to target the subunits D, H, and E, respectively. RNAi suppression of MCO2 causes high mortality in nymphal insects of *P. stali* [24]. Snf7 is a well-known target gene for controlling the western corn rootworm *D. virgifera* [5, 13, 14].

## Detection and sequencing of target genes

Total RNAs were extracted from the whole bodies of adult females approximately 5 days after ecdysis using the RNeasy Mini Kit (Qiagen, Hilden, Germany). Complementary DNAs (cDNAs) were sequenced by Illumina HiSeq 2500 with paired-end 101 bp (Macrogen Japan Corp., Kyoto, Japan), and the generated raw reads (accession numbers DRR118506– DRR118507) were analyzed as previously described [23]. The assembled sequences were subjected to BLASTx database searches, by which multicopper oxidase 2 (*PsMCO2*; accession number LC495720), vacuolar ATPase (*PsvATPase*; accession numbers LC581883–LC581885), inhibitor of apoptosis (*PsIAP*; accession number LC581886), Snf7 (*PsSnf7*; accession number LC581887), dsRNase (*PsdsRNase*; accession number LC581888), and fatty acid synthase (*Psfasn*; accession number LC581889) were identified. On the basis of each predicted sequence, we designed primer pairs for quantitative RT-PCR and RNAi (S1 and S2 Tables).

## dsRNA synthesis

Template preparation for dsRNA synthesis was performed by PCR using the designed primers (S2 Table) in combination with the T7 promoter sequence at the 5′ end. PCR was done by 35 cycles of 10 sec at 96˚C, 30 sec at 55˚C and 1 min at 72˚C using Takara EX taq (Takara Bio, Shiga, Japan). PCR products were purified using the QIAquick Gel Extraction kit (Qiagen, Hilden, Germany) and subjected to transcription into dsRNA using RiboMAX Large Scale RNA Production Systems (Promega, Madison, WI, USA). pEGFP-L1 (Takara Bio) was used as template of enhanced green fluorescent protein (EGFP). The concentrations of dsRNA were estimated by Nanodrop Lite (Thermo Fisher Scientific, Wilmington, DE, USA).

## RNAi knockdown of target genes through injection

For suppression of the mRNA levels of target genes by injection, newly molted fifth (last) instar nymphs were injected with dsRNA solution (approximately 3 μL) into the ventral septum between the thoracic and abdominal segments.

## RNAi knockdown of target genes through feeding

For RNAi by feeding, fifth-instar or first-instar nymphs were reared under the conditions similar to those for the stock colony as cited above, except for the drinking water ingredients. Newly molted fifth-instar nymphs were supplied with an absorbent cotton (3 × 3 cm; Iwatsuki, Tokyo, Japan) soaked with 6 mL of dsRNA solution (100 or 1000 ng/μL), and newly hatched

first-instar nymphs were supplied with one-eighth cut absorbent cotton soaked with 800 μL of dsRNA solution (100, 1000, or 5000 ng/μL) in a plastic dish. During the experiments, the absorbent cotton soaked with dsRNA solution was kept being available for *P. stali*.

In sap-sucking insects, RNAi efficiency has often been tested by plant-mediated dsRNA delivery [41–43]. Notably, a previous study on the brown marmorated stinkbug *Halyomorpha halys* belonging to the Pentatomidae (the same family as *P. stali*) showed that nymphs fed with green bean pods immersed in dsRNA solution showed effective suppression of the target gene expression [44]. In this study, we prepared dsRNA-treated green bean pods according to the previous study with some modifications. Commercially available green bean pods (*Phaseolus vulgaris*) were washed with 0.2% sodium hypochlorite and then washed three times with water. The bean pods were trimmed from the calyx end to a total length of 7.5 cm and immersed in a 1.6 mL of 67 ng/μL dsRNA solution. The immersed green bean pods were fed to fifth-instar nymphs of *P. stali*. To minimize evaporation of the dsRNA solution, we sealed the gap between the bean pods and solution with parafilm (Bemis Company, Oshkosh, WI, USA). The bean pods were replaced every three days, on which the fifth-instar nymphs were reared until adult emergence.

## Quantification of gene expression

To estimate RNAi efficiency, total RNA samples were extracted from the whole bodies of first- or fifth-instar nymphs and reverse-transcribed into cDNA using the High-Capacity cDNA Reverse Transcription Kit (Thermo Fisher Scientific, Waltham, MA, USA). Quantitative RT-PCR of the target genes was conducted using LightCycler 480 or LightCycler 96 with LightCycler 480 SYBR Green Master (Roche Diagnostics, Basel, Switzerland). The gene expression levels were estimated by absolute quantification, and relative expression levels were calculated to ribosomal protein L32 (rpL32) gene. Standard curves were generated using 7 steps of ten-fold serial dilutions of each sequence inserted into the plasmid pGEM-T (Promega, Madison, WI, USA). We confirmed that melting curves in each gene showed similar patterns between samples. We also confirmed that Ct value of rpL32 was similar irrespective of samples/treatments. The mean expression level of the control treatment with dsRNA of EGFP was designated as 1.0, and all expression levels were normalized to the control level.

## Results and discussion

### Effects of dsRNA injection

RNAi of *PsMCO2*, *PsvATPase*, *PsIAP*, and *PsSnf7* by injection with 300 ng dsRNA resulted in high mortality of the injected fifth-instar nymphs, although the mortality and the timing of death varied among the different genes. Nymphs injected with either *PsMCO2* dsRNA or *PsIAP* dsRNA never reached adulthood (Fig 1A). Injection with *PsvATPase* dsRNA led to high mortality irrespective of the targeted subunits. As demonstrated in our previous study [24], the nymphs injected with *PsMCO2* dsRNA died during molting without reaching adulthood. Similar to *PsMCO2*, RNAi of *PsvATPase* induced nymphal mortality during molting. By contrast, the nymphs injected with *PsIAP* dsRNA died 2 to 3 days after injection, and those injected with *PsSnf7* dsRNA died 3 to 4 days after injection, with no relation to the molting events (Fig 1B).

Quantitative RT-PCR confirmed that RNAi by injection efficiently worked for the six genes. Even the least effective gene among the six, *PsMCO2*, exhibited over 80% reduction upon injection with *PsMCO2* dsRNA compared with the control injected with EGFP dsRNA (Fig 1C). The other genes exhibited much higher reduction: 98.0% for *PsvATPase* subunit D,

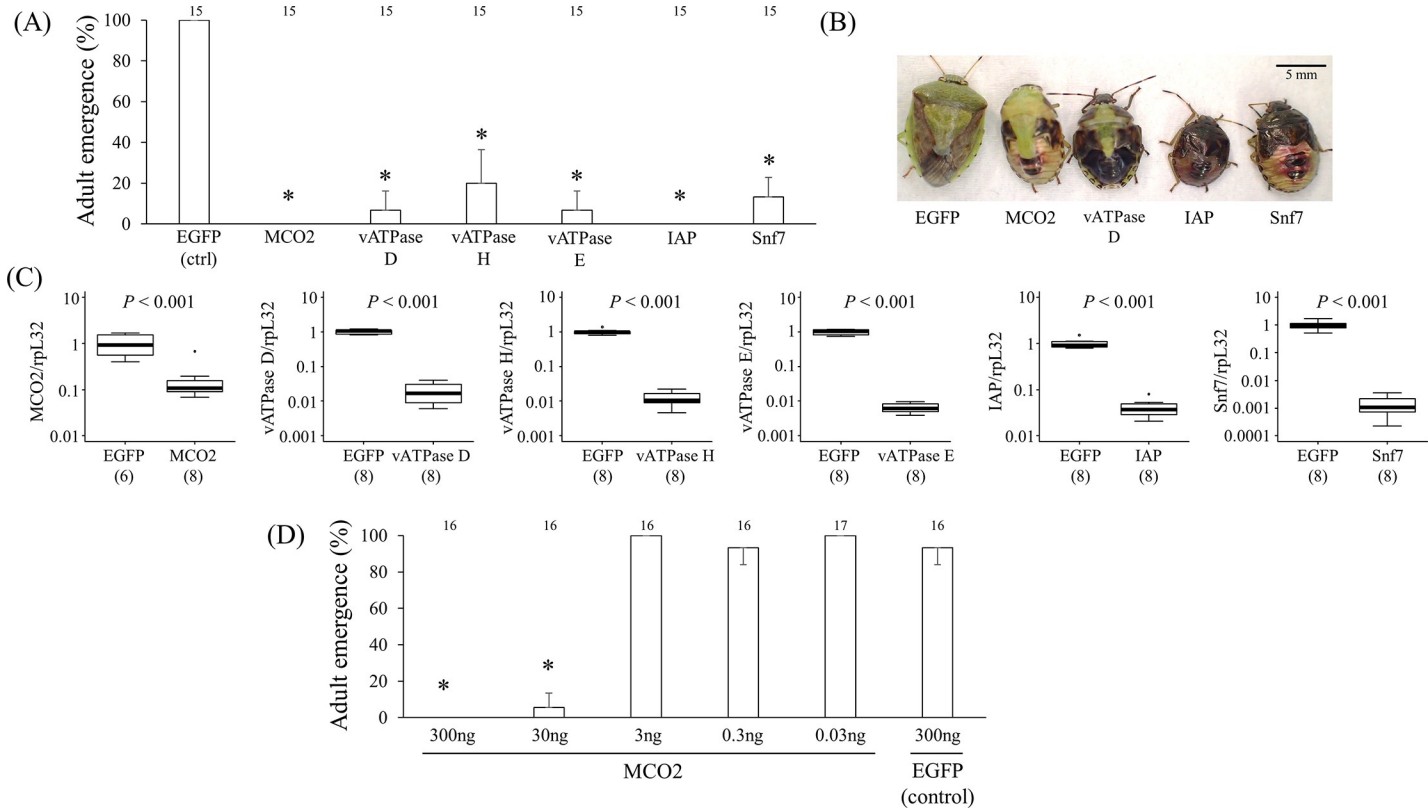

**Fig 1. Effects of RNAi knockdown on fifth-instar nymphs.** (A) Adult emergence rates of fifth-instar nymphs injected with dsRNA against each target gene. Newly molted fifth-instar nymphs were injected with 3 μL of dsRNA at a concentration of 100 ng/μL. Adult emergence rates among 5 nymphs were determined. The experiment was performed in three parallels and the mean adult emergence rates with standard deviations are shown. Numbers above bars show total sample sizes. Statistically significant differences compared with EGFP injection is shown by asterisks (t-test: $P < 0.001$). (B) Normal adult and typical carcasses due to RNAi of *PsMCO2*, *PsvATPase* subunit D, *PsIAP* and *PsSnf7*. Fifth-instar nymphs injected with dsRNA of *PsMCO2* and *PsvATPase* died during molting, whereas nymphs injected with dsRNA of *PsIAP* and *PsSnf7* died independent of the molting events. Scale bar shows 5 mm. (C) Silencing efficiency of RNAi by injection of 300 ng dsRNA into fifth-instar nymphs. RNA was extracted from nymphs 2 days after injection, and the expression level of each gene was quantified. The numbers in parenthesis are sample sizes. Statistically significant differences were evaluated by t-test. The mean expression level of the control treatment (EGFP) was designated as 1.0. (D) Effects of different concentrations of dsRNA on RNAi in fifth-instar nymphs, in which dsRNA targeting *PsMCO2* or EGFP was injected. Newly molted fifth-instar nymphs were injected with 3 μL of dsRNA solution at different concentrations (100, 10, 1, 0.1, or 0.01 ng/μL). Nymphs injected with 30 ng or more dsRNA showed inhibition of cuticle sclerotization and pigmentation. Experiments were performed in three parallels, and the total numbers of sample sizes are shown above the bars. Statistically significant differences compared with EGFP injection is shown by asterisks (t-test: $P < 0.001$).

98.8% for *PsvATPase* subunit H, 99.3% for *PsvATPase* subunit E, 95.9% for *PsIAP*, and 99.9% for *PsSnf7*. Taken together with the previous studies [23, 24], the silencing efficiency in *P. stali* is relatively high among hemipteran insects [33] and similar to the brown marmorated stink-bug *H. halys* [45].

The knockdown of *PsMCO2* resulted in physical defects in cuticle sclerotization and pigmentation. When different concentrations of dsRNA targeting *PsMCO2* were injected, 300 and 30 ng of *PsMCO2* dsRNA caused clear phenotypes in fifth-instar nymphs. Injection of 300 ng dsRNA killed all of the nymphs during molting, and injection of 30 ng dsRNA killed most of them during molting. The one nymph that reached adulthood remained soft and unpigmented, could not feed, and eventually died (Fig 1D). Nymphs injected with 3 ng or lower concentrations of dsRNA did not show any obvious phenotypes in cuticle sclerotization or pigmentation, although accidental death was sometimes observed. These results demonstrate that 30 ng of dsRNA may be a requisite dose to suppress gene expression in fifth-instar nymphs (body weight ~100 mg).

## Effects of orally-delivered dsRNA on fifth-instar nymphs

When 100 or 1000 ng/μL of *PsMCO2* dsRNA solution was supplied to fifth-instar nymphs instead of water, no nymphs showed obvious phenotypes (n = 17 for 100 ng/μL dsRNA solution; n = 18 for 1000 ng/μL dsRNA solution).

We next assessed the effectiveness of plant-mediated RNAi (Fig 2A). When fifth-instar nymphs were fed with green bean pods, around 60%–70% of the nymphs reached adulthood (Fig 2B and 2C). While the survival rates of *P. stali* on green bean pods were relatively low compared with stock colonies fed with raw peanuts and dried soybeans, the plant-mediated 67 ng/μL of dsRNA administration targeting *PsvATPase* and *PsIAP* exhibited little effects on survival of the insects. Quantitative RT-PCR suggested that the plant-mediated RNAi might be effective for *PsvATPase* (approximately 27% reduction; $P = 0.01$ by t-test) but not for *PsIAP*

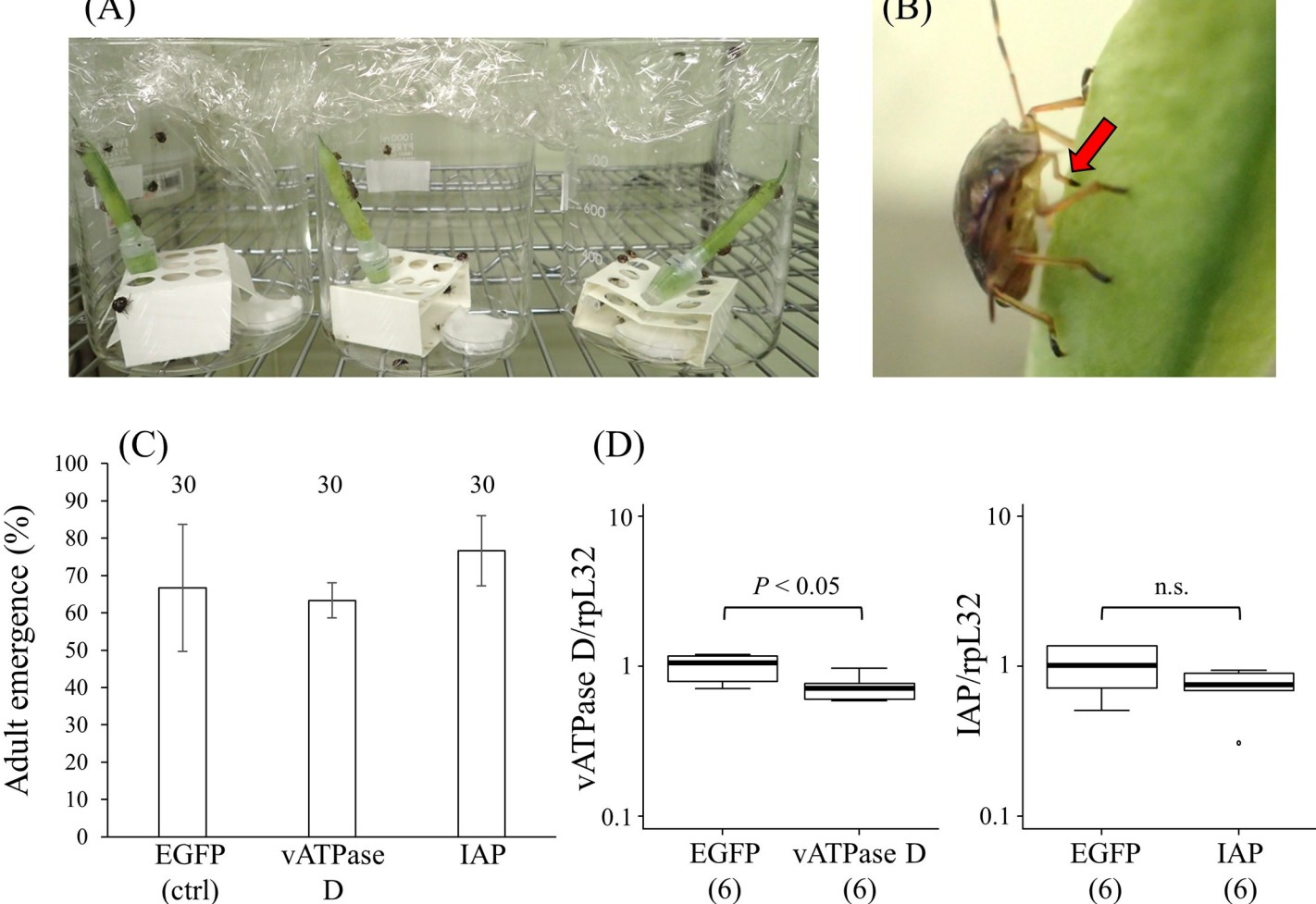

**Fig 2. RNAi by feeding through green bean pods.** (A) Rearing devices with green bean pods. In each 1000 ml beaker, a bean pod was supported to stand with a small paper box and immersed in dsRNA solution with absorbent cotton, on which newly molted fifth-instar nymphs were reared. (B) A fifth-instar nymph feeding on a green bean pod. Red arrow indicates the stylet inserted into the pod. (C) Adult emergence rates of fifth-instar nymphs fed on green bean pods immersed in dsRNA solution. Mean adult emergence rates with standard deviations are shown. Adult emergence rates among 10 nymphs were determined. The experiment was performed in three parallels and the mean adult emergence rates with standard deviations are shown. Numbers above bars show total sample sizes. No statistically significant differences were observed ($P > 0.05$, t-test). (D) Gene silencing efficiency of orally-delivered dsRNA through green bean pods. Total RNAs were extracted 4 days after treatment of fifth-instar nymphs, and the relative expression level of *PsvATPase* subunit D or *PsIAP* to *PsrpL32* was calculated by quantitative RT-PCR. The numbers in parentheses are sample sizes. Statistically significant differences were evaluated by t-test.

(Fig 2D). Compared with the previous study on *H. halys* [44], the plant-mediated RNAi showed lower effectiveness in *P. stali*, although the target genes are different between these studies. The previous study demonstrated that green bean pods immersed in 67 ng/µL of juvenile hormone acid O-methyltransferase (JHAMT) dsRNA solution reduced the corresponding gene expression in the insects by over 70% and green bean pods immersed in 17 ng/µL of vitellogenin dsRNA solution reduced the corresponding gene expression in the insects by over 50%, although no phenotypic descriptions were provided.

### Effects of orally-delivered dsRNA on first-instar nymphs

After hatching, all newborn first-instar nymphs of *P. stali* actively imbibed water without feeding, by which the nymphs became larger in size (Fig 3A), gained an additional body weight by around 0.43 mg (Fig 3B), and molted to second instar three or four days later. These observations indicate that the first-instar nymphs consistently ingest about 0.43 µL of water.

The first-instar nymphs fed with dsRNA solution at concentrations of 100 or 1000 ng/µL (estimated dsRNA intake amounts were 43 and 430 ng, respectively) did not show obvious mortality for all the target genes when compared with the nymphs fed with EGFP dsRNA at the same concentrations ($P > 0.05$; t-test; Fig 3C). Considering that 30 ng of dsRNA is

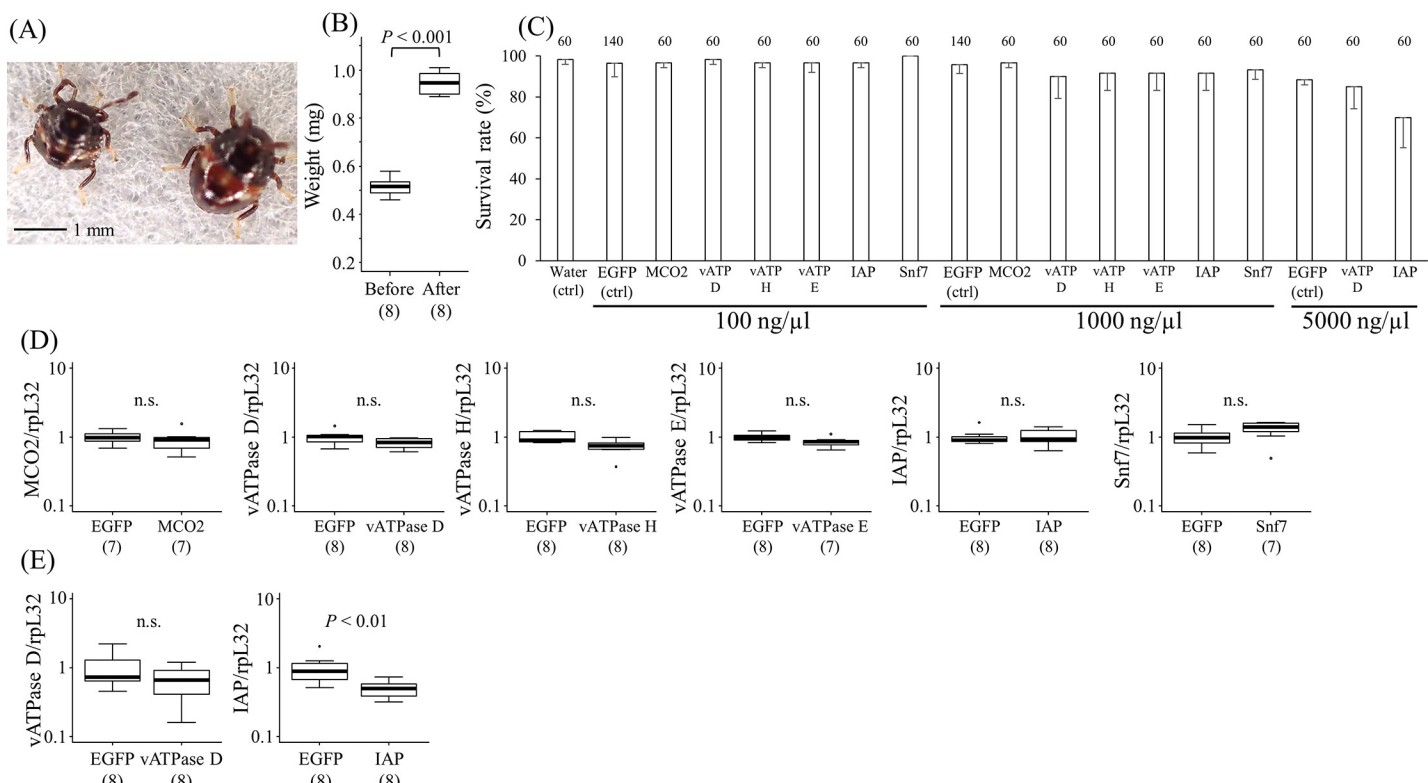

**Fig 3. Effects of orally-delivered dsRNA on the first-instar nymphs.** (A) First-instar nymphs before (left) and after (right) taking water. (B) The weights of the first-instar nymphs before and after taking water. Samples were taken from four separate egg masses. A statistically significant average weight gain of 0.43 mg was observed after the water intake (t-test; $P < 0.001$). (C) Survival rates of the first-instar nymphs. The first-instar nymphs were reared with dsRNA solutions at concentrations of 100, 1000, or 5000 ng/µL for each target gene. 20 nymphs were tested for each gene and the mean rates of molting to second instar were calculated. The experiments were performed in seven parallels for 100 or 1000 ng/µL of EGFP and three parallels for 5000 ng/µL of EGFP and any concentrations of the other genes. Numbers above bars show total sample sizes. (D) Silencing efficiency of orally-delivered dsRNA at 1000 ng/µL in the first-instar nymphs. Statistical analyses were performed by t-test with Bonferroni correction. (E) Silencing efficiency of orally-delivered dsRNA at 5000 ng/µL in the first-instar nymphs. RNAs were extracted from the insects 3 days after providing dsRNA solutions, and the expression level of each gene was quantified. The numbers in parentheses are sample sizes. Statistically significant differences were evaluated by t-test.

sufficient for suppressing the gene expression when injected into fifth-instar nymphs (Fig 1D), *P. stali* appears to exhibit very low sensitivity to orally-delivered dsRNA.

In the first-instar nymphs fed with 1000 ng/μL dsRNA solution, no significant differences were observed in the expression levels of any of the genes tested (t-test with Bonferroni correction; Fig 3D). At 5000 ng/μL, RNAi of *PsIAP* showed slightly higher mortality compared with the control (5000 ng/μL of EGFP dsRNA) (*P* = 0.08; t-test; Fig 3C), although RNAi of *PsvAT-Pase* subunit D exhibited no obvious lethality. In agreement with this, the nymphs fed with 5000 ng/μL *PsIAP* dsRNA (estimated intake, 2,150 ng) showed an around 50% decrease in the gene expression but not in the case of *PsvATPase* subunit D (Fig 3E). The observations that RNAi of *PsIAP* was weakly effective at the dose of 5000 ng/μL but not at 1000 ng/μL indicate that the sensitivity to orally-delivered dsRNA for this target gene is very low in *P. stali*. Comparing with the previous study in the southern green stinkbug *Nezara viridula*, which showed that orally-administrated 3000 ng/μL dsRNA of vATPase suppresses gene expression and leads to high mortality [46], *P. stali* showed lower sensitivity to orally-delivered dsRNA. In the Asian citrus psyllid *Diaphorina citri*, 50% reduction of IAP expression by RNAi was reported to cause high mortality [41], which may be attributable to different sensitivities between the insect species.

Previous studies have suggested the possibility that, in such hemipteran insects as *A. pisum*, *Myzus persicae*, *Lygus lineolaris* and *Bemisia tabaci*, dsRNase in the midgut or saliva may be responsible for the low sensitivity of these insects to RNAi by orally-delivered dsRNA [20, 47–49]. In the oriental fruit fly *Bactrocera dorsalis*, another blocker of RNAi, fasn, was identified [50]. In an attempt to improve the RNAi efficiency, we conducted RNAi experiments in which dsRNA against these potential RNAi blocker genes was simultaneously administrated orally with dsRNA against the target genes. When 1000 ng/μL dsRNA of either *PsdsRNase* or *Psfasn* was orally administrated to first-instar nymphs of *P. stali* together with 1000 ng/μL dsRNA of either *PsvATPase* subunit D or *PsIAP*, no effects on the insect mortality were detected (Fig 4). Possible explanations as to why no obvious effects of *PsdsRNase* RNAi were observed in *P. stali* are (1) low sensitivity to RNAi with orally-delivered dsRNA of *PsdsRNase* in *P. stali*, (2) degradation of *PsdsRNase* dsRNA by dsRNase in the intestine of *P. stali* before cellular uptake, or (3) possible effects of other nucleases present in the intestine of *P. stali*. In the desert locust *Schistocerca gregaria*, it was reported that RNAi of dsRNases did not improve the sensitivity to RNAi [51]. Fasn plays an important role in the loss of endocytic ability for dsRNA in *B. dorsalis* [50], but may not in *P. stali*. Other possible explanation causing the low sensitivity to feeding RNAi may be factors blocking cellular uptake and endosomal escape of dsRNA [52].

## Conclusions

As previously shown, the brown-winged green stinkbug *P. stali* exhibits high RNAi sensitivity to injected dsRNA. Here, we demonstrate by RNAi experiments that *PsMCO2*, *PsvATPase*, *PsIAP*, and *PsSnf7* are essential for survival of *P. stali*, suggesting the possibility that these genes may potentially be useful for controlling the pest stinkbug. Meanwhile, *P. stali* shows very limited sensitivity to orally-delivered dsRNA, where only an extremely high concentration of dsRNA (5000 ng/μL) can induce weak phenotypes depending on the target gene. In the stinkbugs *H. halys* and *N. viridula* belonging to the same family Pentatomidae, by contrast, orally-delivered dsRNA effectively causes gene silencing. These results indicate that sensitivity to orally-delivered dsRNA may vary among closely-related species, which highlight the importance of optimizing dsRNA administration protocol for each pest insect species. Attempts to improve the efficiency of orally-delivered RNAi in *P. stali* by, for example, supplementation of cationic nanoparticles [53, 54] should be conducted in future studies.

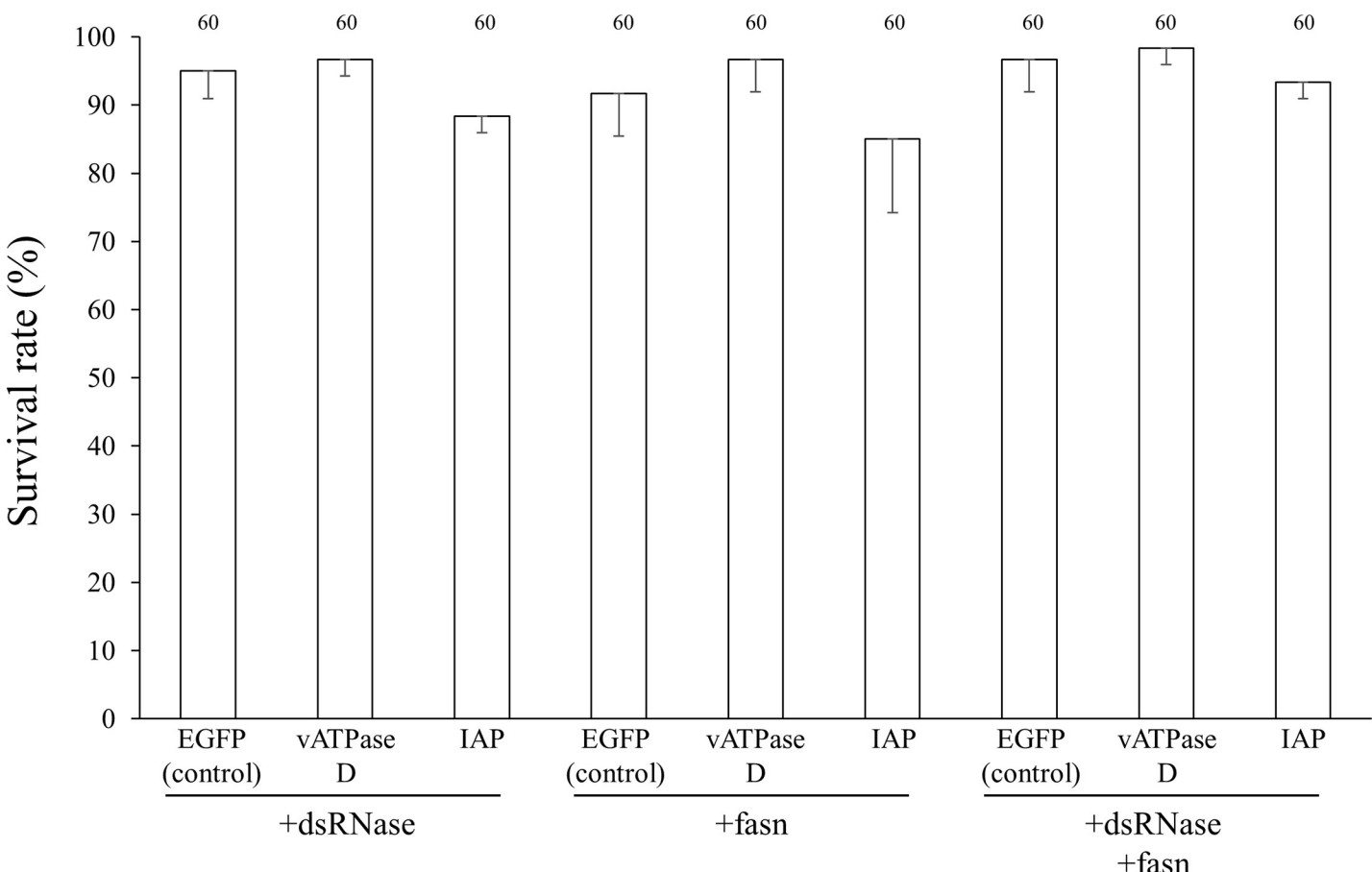

**Fig 4. Effects of simultaneous feeding of dsRNA against potential RNAi blocker genes on RNAi efficiency of target genes.** Survival rates of first-instar nymphs. The first-instar nymphs were reared with dsRNA solutions in which dsRNA against each gene is present at a concentration of 1000 ng/μL. In comparison with the control treatment (EGFP), no statistically significant differences in survival rates were detected by t-test. 20 nymphs were tested for each gene and the mean rates of molting to second instar were calculated. The experiment was performed in three parallels. Numbers above bars show total sample sizes.

## Supporting information

**S1 Table. Primer sequences for quantitative RT-PCR.**
(XLSX)

**S2 Table. Primer sequences for RNAi.**
(XLSX)

## Acknowledgments

We thank Masae Takashima for maintaining stinkbug colonies and technical assistance.

## Author Contributions

**Conceptualization:** Yudai Nishide, Yoshiaki Tanaka.

**Data curation:** Yudai Nishide.

**Formal analysis:** Yudai Nishide, Kakeru Yokoi, Akiya Jouraku, Ryo Futahashi.

**Funding acquisition:** Yudai Nishide, Takema Fukatsu.

**Investigation:** Yudai Nishide.

**Methodology:** Yudai Nishide.

**Project administration:** Yudai Nishide, Takema Fukatsu.

**Resources:** Yudai Nishide.

**Supervision:** Yudai Nishide, Daisuke Kageyama, Yoshiaki Tanaka, Ryo Futahashi, Takema Fukatsu.

**Validation:** Yudai Nishide.

**Visualization:** Yudai Nishide, Daisuke Kageyama.

**Writing – original draft:** Yudai Nishide.

**Writing – review & editing:** Daisuke Kageyama, Yoshiaki Tanaka, Ryo Futahashi, Takema Fukatsu.

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
