## [Decision Letter · Decision Letter 0]

11 Nov 2020

PONE-D-20-30716

Effectiveness of orally-delivered double-stranded RNA on gene silencing in the stinkbug Plautia stali

PLOS ONE

Dear Dr. Nishide,

Thank you for submitting your manuscript to PLOS ONE. After careful consideration, we feel that it has merit but does not fully meet PLOS ONE’s publication criteria as it currently stands. Therefore, we invite you to submit a revised version of the manuscript that addresses the points raised during the review process.

Both reviewers have raised concerns about deficiencies in the Materials and Methods, and these must be thoroughly addressed in your revision. Comments regarding English language quality should be addressed as well. Note that comments made regarding novelty are not relevant to PLOS ONE. 

We look forward to receiving your revised manuscript.

Kind regards,

Zach N Adelman

Academic Editor

PLOS ONE

Journal Requirements:

Reviewers' comments:

Reviewer's Responses to Questions

**Comments to the Author**

1. Is the manuscript technically sound, and do the data support the conclusions?

Reviewer #1: Partly

Reviewer #2: Yes

2. Has the statistical analysis been performed appropriately and rigorously? 

Reviewer #1: Yes

Reviewer #2: Yes

3. Have the authors made all data underlying the findings in their manuscript fully available?

Reviewer #1: Yes

Reviewer #2: Yes

4. Is the manuscript presented in an intelligible fashion and written in standard English?

Reviewer #1: No

Reviewer #2: Yes

5. Review Comments to the Author

Reviewer #1: Dear Editor

The manuscript entitled "Effectiveness of orally-delivered double-stranded RNA on gene silencing in the stinkbug Plautia stali” describes a RNAi study by oral-delivered and microinjection dsRNA of several genes in order to evaluate Plautia stali response. This study has interesting data. However, I have several concerns about how the manuscript is written, and lack some comments that I'd like the authors to consider for the possible improvement of the manuscript, outlined below.

Major comments:

1.The English in the present manuscript is not of publication quality and require major improvement.

2. Abstract. Is recommended don't include too many details

3. Description of microinjection experiments should be more detailed in materials and methods, including sample size, treatments, dsRNA concentrations. Most data included in figures references should be included in Materials and Methods section.

4. In methods section, it should include the sample size used to obtain microinjection and feeding assays data. The values only are mentioned in figure references. Only 5 nymphs were used in order to evaluate adult emergence rates, 6-8 nymphs were used in some gene knockdown experiments and some feeding assays. These samples size seem to be too small. Similar published papers used between 20-30 insect in each treatment. Also, dsRNA GFP synthesis (control) is not mentioned in materials and methods.

5. During qPCR experiments analysis only one gene is used as reference gene. How de standard curve was performed? Dilutions? The information about de reference gene is lacked.

6. The exposition time to dsRNA or time of insect feeding with dsRNA is an important factor for successful induction of gene silencing. In this study, the authors performed a feeding assay on first-instar nymphs to second-instar nymphs (using with dsRNA solutions at different concentrations). In other assay, the authors evaluated the dsRNA effect from fifth-instar nymph to adults by insect feeding through green bean pods. Is possible evaluate the continuous feeding with dsRNA from first-instar nymphs to adults?

Reviewer #2: The main objective of this study was to compare the efficiency of RNAi by injection vs feeding in the brown-winged green stinkbug Plautia stali (Hemiptera: Pentatomidae). Authors concluded that P. stali shows high sensitivity to RNAi with injected dsRNA but, unlike the allied pest stinkbugs Halyomorpha halys and Nezara viridula, very low sensitivity to RNAi with orally delivered dsRNA. Overall, this article is well presented and organized. The following suggestion might strengthen the manuscript and make it suitable for publication in PlosOne:

1. It would be important to highlight the novelty of the study in the introduction and conclusions. Are any of the target genes have been tested before by other researches? RNAi background on Plautia stali is missing.

2. Authors should explain why the concentrations of dsRNA tested through injection vs feeding were different.

3. The method section is missing details. A separate section should be created for the feeding assays and injection.

4. It is not clear whether the insects were starving before exposing the insects to diets supplemented with dsRNA.

5. Authors should consider the use of cationic nanoparticles to increase the effectiveness of oral delivery.

6. PLOS authors have the option to publish the peer review history of their article (what does this mean?). If published, this will include your full peer review and any attached files.

Reviewer #1: No

Reviewer #2: No

---

## [Author Response · Author response to Decision Letter 0]

26 Nov 2020

Reviewer #1: Dear Editor

The manuscript entitled "Effectiveness of orally-delivered double-stranded RNA on gene silencing in the stinkbug Plautia stali” describes a RNAi study by oral-delivered and microinjection dsRNA of several genes in order to evaluate Plautia stali response. This study has interesting data. However, I have several concerns about how the manuscript is written, and lack some comments that I'd like the authors to consider for the possible improvement of the manuscript, outlined below.

Response: We greatly appreciate your time and effort for reviewing our manuscript.

Major comments:

> Abstract. Is recommended don't include too many details

Response: Done.

> Description of microinjection experiments should be more detailed in materials and methods, including sample size, treatments, dsRNA concentrations. Most data included in figures references should be included in Materials and Methods section.

Response: Because the methods and sample sizes are different among the experiments, we described all the information on the figures and in the captions, which must be more reader-friendly we believe.

> In methods section, it should include the sample size used to obtain microinjection and feeding assays data. The values only are mentioned in figure references. Only 5 nymphs were used in order to evaluate adult emergence rates, 6-8 nymphs were used in some gene knockdown experiments and some feeding assays. These samples size seem to be too small. Similar published papers used between 20-30 insect in each treatment. Also, dsRNA GFP synthesis (control) is not mentioned in materials and methods.

Response: We performed the experiments with no less than 3 replicates using no less than 5 nymphs. To clarify this, we added total numbers of samples in the revised manuscript. Also the description of EGFP dsRNA synthesis was added (L. 131-133).

5. During qPCR experiments analysis only one gene is used as reference gene. How de standard curve was performed? Dilutions? The information about de reference gene is lacked.

Response: These details were added in the revised manuscript (L. 173-176).

6. The exposition time to dsRNA or time of insect feeding with dsRNA is an important factor for successful induction of gene silencing. In this study, the authors performed a feeding assay on first-instar nymphs to second-instar nymphs (using with dsRNA solutions at different concentrations). In other assay, the authors evaluated the dsRNA effect from fifth-instar nymph to adults by insect feeding through green bean pods. Is possible evaluate the continuous feeding with dsRNA from first-instar nymphs to adults?

Response: In the revised manuscript, the exposure time was explicitly described (L. 147-148).

Indeed it is possible to rear the insects with dsRNA-supplemented food from first-instar to adulthood, but the experiment will cost too much, particularly when high-concentrated dsRNA solution is used.

Reviewer #2: The main objective of this study was to compare the efficiency of RNAi by injection vs feeding in the brown-winged green stinkbug Plautia stali (Hemiptera: Pentatomidae). Authors concluded that P. stali shows high sensitivity to RNAi with injected dsRNA but, unlike the allied pest stinkbugs Halyomorpha halys and Nezara viridula, very low sensitivity to RNAi with orally delivered dsRNA. Overall, this article is well presented and organized. The following suggestion might strengthen the manuscript and make it suitable for publication in PlosOne:

Response: We greatly appreciate your time and effort for reviewing our manuscript.

> It would be important to highlight the novelty of the study in the introduction and conclusions. Are any of the target genes have been tested before by other researches? RNAi background on Plautia stali is missing.

Response: We added relevant information in Introduction (L. 79-80).

> Authors should explain why the concentrations of dsRNA tested through injection vs feeding were different.

Response: As cited in Results and Discussion (L. 237-239), we started both the experiments with the concentration of 100 ng/ul. Then, we performed injection RNAi with lower concentrations and feeding RNAi with higher concentrations, because 100 ng/ul was effective for injection RNAi but not effective for feeding RNAi.

> The method section is missing details. A separate section should be created for the feeding assays and injection.

Response: Done.

> It is not clear whether the insects were starving before exposing the insects to diets supplemented with dsRNA.

Response: We added the information (L. 143 and 145). Special starving treatment was not done.

> Authors should consider the use of cationic nanoparticles to increase the effectiveness of oral delivery.

Response: We mentioned the potential utility of nanoparticles in discussion (L. 361-363).

---

## [Decision Letter · Decision Letter 1]

22 Dec 2020

Effectiveness of orally-delivered double-stranded RNA on gene silencing in the stinkbug Plautia stali

PONE-D-20-30716R1

Dear Dr. Nishide,

We’re pleased to inform you that your manuscript has been judged scientifically suitable for publication and will be formally accepted for publication once it meets all outstanding technical requirements.

Kind regards,

Zach N Adelman

Academic Editor

PLOS ONE

Additional Editor Comments (optional):

While one reviewer requested a few small additional edits to the manuscript, these were based on increasing perceived impact, which cannot be used as a criteria for acceptance. 

Reviewers' comments:

Reviewer's Responses to Questions

**Comments to the Author**

1. If the authors have adequately addressed your comments raised in a previous round of review and you feel that this manuscript is now acceptable for publication, you may indicate that here to bypass the “Comments to the Author” section, enter your conflict of interest statement in the “Confidential to Editor” section, and submit your "Accept" recommendation.

Reviewer #1: All comments have been addressed

Reviewer #2: (No Response)

2. Is the manuscript technically sound, and do the data support the conclusions?

Reviewer #1: Yes

Reviewer #2: Yes

3. Has the statistical analysis been performed appropriately and rigorously? 

Reviewer #1: Yes

Reviewer #2: Yes

4. Have the authors made all data underlying the findings in their manuscript fully available?

Reviewer #1: Yes

Reviewer #2: Yes

5. Is the manuscript presented in an intelligible fashion and written in standard English?

Reviewer #1: Yes

Reviewer #2: No

6. Review Comments to the Author

Reviewer #1: Dear Editor

The authors have made valuable improvements in the manuscript and most of the problems that I pointed were solved. In my opinion, it is now acceptable for publication.

Reviewer #2: Question one from reviewer 2 was partially answered. The authors only added one line and two references. Explaining how this study is different from similar articles showing dsRNA silencing effects in Plautia stali will draw more attention. Also, it would be desirable to add more information about the target genes.

7. PLOS authors have the option to publish the peer review history of their article (what does this mean?). If published, this will include your full peer review and any attached files.

Reviewer #1: No

Reviewer #2: No

---

## [Editor Report · Acceptance letter]

6 Jan 2021

PONE-D-20-30716R1 

Effectiveness of orally-delivered double-stranded RNA on gene silencing in the stinkbug *Plautia stali*

Dear Dr. Nishide:

I'm pleased to inform you that your manuscript has been deemed suitable for publication in PLOS ONE. Congratulations! Your manuscript is now with our production department. 

Kind regards, 

on behalf of

Dr. Zach N Adelman 

Academic Editor

PLOS ONE